# Crack Inhibition and Performance Modification of NiCoCr-Based Superalloy with Y_2_O_3_ Nanoparticles by Laser Metal Deposition

**DOI:** 10.3390/ma16103616

**Published:** 2023-05-09

**Authors:** Xiaodong Li, Jiaxin Du, Jijin Xu, Shuai Wang, Mengling Shen, Chuanhai Jiang

**Affiliations:** 1School of Materials Science and Engineering, Shanghai Jiao Tong University, Shanghai 200240, China; qinnashou@sjtu.edu.cn (X.L.); djx19981216@situ.edu.cn (J.D.); wangshuai17@sjtu.edu.cn (S.W.); 2Shanghai Key Laboratory of Materials Laser Processing and Modification, School of Materials Science and Engineering, Shanghai Jiao Tong University, Shanghai 200240, China; 3China Nuclear Industry 23 Construction Co., Ltd., Beijing 101300, China; shenmengling@126.com; 4Science and Technology on Green Construction Laboratory, Beijing 101300, China

**Keywords:** NiCoCr-based superalloy, Y_2_O_3_ nanoparticles, hot cracks, laser metal deposition

## Abstract

A new precipitation strengthening NiCoCr-based superalloy with favorable mechanical performance and corrosion resistance was designed for ultra-supercritical power generation equipment. The degradation of mechanical properties and steam corrosion at high temperatures put forward higher requirements for alternative alloy materials; however, when the superalloy is processed to form complex shaped components through advanced additive manufacturing techniques such as laser metal deposition (LMD), hot cracks are prone to appear. This study proposed that microcracks in LMD alloys could be alleviated with powder decorated by Y_2_O_3_ nanoparticles. The results show that adding 0.5 wt.% Y_2_O_3_ can refine grains significantly. The increase in grain boundaries makes the residual thermal stress more uniform to reduces the risk of hot cracking. In addition, the addition of Y_2_O_3_ nanoparticles enhanced the ultimate tensile strength of the superalloy at room temperature by 18.3% compared to original superalloy. The corrosion resistance was also improved with 0.5 wt.% Y_2_O_3_, which was attributed to the reduction of defects and the addition of inert nanoparticles.

## 1. Introduction

A new kind of NiCoCr-based superalloy has been developed for ultra-supercritical power equipment at temperatures above 650 °C [1]. The degradation of mechanical properties and steam corrosion at high temperature put forward higher requirements for alternative alloy materials. As a precipitated strengthening alloy, it can maintain considerable mechanical properties at high temperature. In addition, the high content of Cr also gives the alloy excellent oxidation and corrosion resistance [2]. Therefore, it is considered a suitable candidate material for ultra-supercritical power generation equipment.

In recent years, laser metal deposition (LMD) has been employed for accurate repair and rapid manufacturing of Ni-based superalloy components, as it offers rapid heating and cooling, low dilution rates, and strong metallurgical bonding of the base metal with the cladded layer [3]; however, LMD delivers near net-shaped products through layer-by-layer manufacturing with each cladded layer having a different thermal history, resulting in significant changes within their microstructures along the growth direction. Meanwhile, the extremely high rate of cooling during LMD of up to 10^6^ K/s leads to high residual stress. Additionally, Al and Ti are the most common alloy elements used for deposition-strengthened alloys, and upon precipitation of the strengthening phase, γ′ (Ni_3_(Al, Ti)) is likely to be observed [4]. Typically, the alloy cannot be welded if the overall content of Al and Ti exceeds 6 wt.% [5]. For these reasons, hot cracks are heavily formed in the NiCoCr-based superalloys prepared using the LMD technique.

In order to prevent the occurrence of hot cracks, exploring the mechanism behind crack formation within the deposited layers of NiCoCr-based superalloys is of great significance [6,7]; however, the mechanisms of hot cracking are influenced tremendously by the alloy compositions and processing methods. Therefore, previous studies devoted to both the microstructure as well as the crack formation in NiCoCr-based superalloys prepared by LMD have so far been rather inconclusive. The study undertaken by Chen et al. [8] claimed that the precipitation of γ′ particles from the austenite γ matrix was not viable in the deposited IN738LC alloy owing to the intrinsic rapid cooling in the LMD process. Hence, post-weld heat treatment plays a key role in acquisition of the required thermomechanical properties by precipitation of the γ′ particles; however, Xu et al. [9] reported that 100 nm γ′ particles were, indeed, present in the γ matrix, and that the generated cracks were induced by a eutectic reaction of γ + γ′ or by liquefaction of borides along the grain boundaries. This conclusion was drawn on the basis of laser cladding of the deposited IN738LC samples in cases of powder-free feeding. In selective laser melting (SLM) of the IN738LC alloy, cracks were induced by consolidation, and Zr segregation played a key role in the initiation of these cracks [10]. Interestingly, however, Ramakrishnan et al. [11] found that cracks formed in the IN738 alloy during LMD were induced by both consolidation and liquefaction, which can be attributed to the low melting point eutectic reaction of γ + γ′, carbides and borides. Additionally, Zhang et al. [12] investigated the deposited layers of IN738 alloy on a directionally solidified superalloy matrix. Herein, cracks of five different categories were found, and it was revealed that these cracks were initiated at locally liquefied grain boundaries of the matrix, which then propagated towards the deposition layer.

It has previously been demonstrated that addition of rare-earth oxides to Ni-based superalloy powders during manufacturing of laser additives leads to grain refinement and grain boundary purification [13] as well as to enhancement of the overall alloy performance (e.g., strength and ductility) [14]. In this study, Y_2_O_3_ was added into the alloy powder to reduce or eliminate hot cracks initiated during the manufacturing process of the NiCoCr-based superalloy by LMD. Additionally, the effects of Y_2_O_3_ on microstructure of the cladding layers were studied, and the mechanism of Y_2_O_3_ inhibiting liquefaction cracking of Ni-based superalloy was investigated.

## 2. Materials and Methods

The chemical compositions of the proposed NiCoCr-based superalloy powder are summarized in Table 1. As provided in Table 1, the matrix including Ni, Co, and Cr exhibits good thermal resistances and mechanical behavior under high temperature with the addition of Al, Ti, and Nb, which are used as solution-strengthening elements [15,16]. W is also added to enhance wear resistance and generate strengthening effects by causing lattice distortion because of its largest atomic radius [17]. The powder was prepared by the argon blowing method, and their particle sizes were measured to be approximately 50 μm~75 μm.

The complete setup used for carrying out the LMD process in this study is shown in Figure 1. It comprised of a six-axis robot (KUKA, Augsburg, Germany), a coaxial laser cladding head, a laser generator, a powder feeder, a water-cooling system, a robot controller and a process planner.

The particle size of the white spherical Y_2_O_3_ powder used in this study was in the range of 50–100 nm with a purity of 99.9% and a melting point of about 2415 °C. The Y_2_O_3_ powder was added into the NiCoCr-based superalloy powder by employing a ball milling (XQM-04 vertical planetary mill, rotation rate = 300 r/min, duration = 3 h, ball-to-powder ratio = 2:1). The results of SEM imaging and Y-element EDS spectroscopy for the NiCoCr-based superalloy powder containing Y_2_O_3_ are provided in Figure 2a,b, respectively. As can be seen in the figure, Y_2_O_3_ was homogeneously distributed on the surface of the NiCoCr-based superalloy powder after the ball milling process, wherein no significant aggregation was observed, which favors the synthesis of Y_2_O_3_/NiCoCr-based composite powder by LMD.

The optimized processing parameters were taken with laser power of 800 W, scanning speed of 10 mm/s, hatch spacing of 0.8 mm, and a layer thickness of 0.5 mm. All printed samples had dimensions of 20 mm × 10 mm × 3 mm. The samples were sliced by wire-electrode cutting method in a direction perpendicular to the deposition direction. The obtained cross-sectional samples were then ground and polished for metallographic analysis. Tensile samples were taken at a site more than 0.5 mm away from the substrate to counter the effects of the dilution rate. The sample was etched in aqua regia (HNO_3_:HCl = 3:1, volume ratio), and the precipitated phases and microstructures were investigated by using scanning electron microscopy (SEM, VEGA3 TEScan, Brno, Czech Republic). Electropolishing was performed in a 10% perchloric acid/alcohol solution (HClO_4_:C_2_H_6_OH = 1:9, volume ratio) by using an electrochemical station (voltage = 30 V, period = 10 s). Grain orientation was studied using SEM equipped with an electron backscatter diffraction (EBSD) probe.

## 3. Results and Discussion

### 3.1. Initiation of Hot Cracks

The metallographic structure of the multi-layer, multi-pass, NiCoCr-based superalloy prepared by LMD is shown in Figure 3. As can be seen in the figure, several cracks were formed in the deposited layer. Additionally, a majority of these cracks started to develop from the upper-middle part of the deposited layer and propagated upwards along its height direction. The lengths of these cracks were measured to be from 300 μm to 1400 μm. Figure 3b,c illustrate the morphology of dendritic crystals in the crack propagation zone. As observed from the figures, there was a huge difference in orientation between the dendritic crystals on each side of the large cracks, indicating that the cracks had a tendency to initiate and propagate along the large-angle grain boundaries.

Figure 4a shows the crack morphology of the NiCoCr-based superalloy prepared by LMD. The high-angle and low-angle grain boundaries of the crack zone as processed by Channel 5 software are shown in Figure 4b. Herein, green lines represent low-angle grain boundaries (2° < orientation difference < 15°), whereas the high-angle grain boundaries (orientation difference > 15°) are represented by black lines. An analysis of the figure based on this representation clearly suggested that cracks were initiated at the grain boundaries and that they exhibited significant intergranular crack features.

Figure 4c shows the morphology of the formed cracks after erosion in the NiCoCr-based superalloy prepared by LMD. These cracks caused drastic reduction in stability of the grain boundaries, resulting in a severe risk of irreversible damage during service of the deposited layer. A high-resolution SEM image of a crack is provided in Figure 4d. The crack in the SEM image can be seen to have a smooth surface on both sides, which can be attributed to the surface tension of intergranular liquid phase during its solidification. Additionally, traces of plastic deformation were observed, indicating that the crack was induced by the liquefaction process. Hence, it can be concluded that cracks generated in the NiCoCr-based superalloy during LMD were actually induced by liquefaction.

Cracks formed in NiCoCr-based superalloy prepared by LMD can be associated with component liquefaction during the process of osmosis. It is mainly that the second phase particles in the heat-affected zone have a eutectic reaction with the substrate of the deposition layer [18]. Figure 5a provides a SEM image of the cracks induced by liquefaction in the NiCoCr-based superalloy prepared by LMD. Herein, abundant white carbide particles distributed on both sides of the crack can be observed, whereas abundant re-solidified products were found inside the crack. The results of the EDS spectroscopy for these cracks are provided in Figure 5b. Ti aggregation was observed near the crack, suggesting that the white carbide particles on both sides of the crack were indeed MC-type carbides. On the other hand, C aggregation was observed inside the cracks with hardly any traces of Ni, Cr, and Co found within of them. These observations strongly suggested that this crack was induced by intergranular liquefaction of MC-type carbides. The presence of continuously distributed intergranular carbide-based liquefaction products provided paths for propagation of liquefaction-induced cracks, leading to the generation of abundant re-solidified products inside the crack after cooling. Similar to the liquefaction of γ + γ′ eutectics [18], MC-type carbides were exposed to a eutectic reaction with the deposition layer γ matrix once the temperature reached the eutectic temperature. A liquid film was generated at the grain boundaries after liquefaction of the eutectic products that was torn apart by the residual stress, resulting in generation of liquefaction-induced cracks. For the NiCoCr-based superalloy, the initial melting temperature of the MC + γ eutectics was approximately 1086 °C [19], which was significantly lower than the temperature of the molten pool of LMD (2000 °C) [20]. Hence, it clearly suggests that MC + γ eutectic reaction was the dominant cause for initiation of the liquefaction-induced cracks in NiCoCr-based superalloys prepared by LMD.

### 3.2. Effects of Y_2_O_3_ on Hot Cracks

The macroscopic morphologies along the growth direction of the layers prepared by LMD with Y_2_O_3_ weight percentages of 0 wt.% and 0.5 wt.% are shown in Figure 6a,b, respectively. In comparison to the deposited layer without Y_2_O_3_, almost no visible cracks were observed in the deposited layer with 0.5wt.% Y_2_O_3_. Additionally, the overall molten pool was found to be more homogeneous, indicating that the addition of Y_2_O_3_ effectively eliminated the liquefaction-induced cracks, leading to significant improvement in the microstructure, as well as uniformity of the NiCoCr-based superalloy prepared by LMD.

The SEM images of the deposited layers with 0 wt.% and 0.5 wt.% of Y_2_O_3_ prepared by LMD are provided in Figure 6c,d, respectively. It can be seen that both samples exhibited a typical morphology of dendritic crystals in NiCoCr-based superalloys prepared by LMD, suggesting that the addition of Y_2_O_3_ did not affect the morphology of the microstructure. Additionally, abundant indentures were observed at the boundaries of the dendritic crystals, owing to etching or exfoliation of carbides during erosion, as indicated by the arrows in Figure 6c,d. Notably, small spherical precipitates were observed in dendritic crystal grains of the deposited layer with 0.5 wt.% Y_2_O_3_.

The inverse figure pole (IFP) maps obtained from the results of EBSD along the growth direction of the deposited layers with 0 wt.% and 0.5 wt.% Y_2_O_3_ are illustrated in Figure 6e,f, respectively. Obviously, columnar dendritic grains grow continuously along a specific direction and penetrate through multiple layers within the microstructure. This can be attributed to the highly oriented laser-induced temperature gradient in the LMD technique. Especially, for the deposited layer with 0 wt.% Y_2_O_3_, significant growth of the columnar dendritic grains took place, where the average aspect ratio of the grains was found to be approximately 10:1. In comparison, for the deposited layer with 0.5 wt.% Y_2_O_3_, the columnar dendritic grains did not grow as much, and their average aspect ratio was limited to approximately 5:1. The addition of Y_2_O_3_ led to significantly reduced grain size. Previous studies show that finer grains are more vulnerable to rotation and deformation during rapid solidification of molten pool and are more resistant to strains of the molten pool in a semi-solid state, thereby preventing crack initiation and propagation within them during the solidification process [21]. Finer grains allow for effective tuning of the stress distribution during the LMD process [22,23] and provide more complicated paths for crack propagation [24]. The XRD patterns were also compared in Figure 7. Additionally, it can be seen that both samples have typical FCC characteristics. With the addition of Y_2_O_3_, the intensity of grain orientations (200) and (220) are enhanced, which may be related to grain refinement, resulting in a slight increase in anisotropy; however, Because of the low content of Y_2_O_3_ (0.5 wt.%) and the strong tendency of epitaxial growth of NiCoCr-based superalloy, there is no significant difference between the XRD patterns. Therefore, initiation and propagation of cracks posed a more serious challenge in the deposited layer with 0.5 wt.% Y_2_O_3_ in comparison to that in the deposited layer with without Y_2_O_3_.

### 3.3. Effects of Y_2_O_3_ on the Performance of the Deposited Layer

The tensile test results for the NiCoCr-based superalloy as well as the Y_2_O_3_/NiCoCr-based superalloys prepared by the LMD technique are shown in Figure 8. The calculated yield strengths of the two alloys obtained from these results were found to be 681 MPa and 692 MPa, respectively, whereas their tensile strengths were calculated to be 970 MPa and 1148 MPa, respectively (increase by 18.3%). Additionally, their elongations were found to be 6.1% and 14.4%, respectively, which shows an increase by 136% for the alloy with 0.5 wt.% Y_2_O_3_. Their yield ratios were 0.72 and 0.61, respectively. Theoretically, an increase in the yield ratio is reflected as a decrease in the deformation strengthening capability, and a corresponding decrease in the strain hardening index results in an increased risk of brittle failure [25]. According to their tensile curves, the elastic deformation stages of the two samples were highly consistent (stress–strain curves were linear, and the slopes were identical); however, the sample with 0.5 wt.% Y_2_O_3_ exhibited a slightly higher tensile slope and better plasticity at the beginning of the plastic deformation stage. Additionally, the elongation was significantly enhanced, which implied that both the plasticity as well as the toughness of the alloy had improved greatly.

Electrochemical tests were conducted on the deposited layers with 0 wt.% and 0.5 wt.% of Y_2_O_3_. Herein, a saturated calomel electrode was employed as the reference electrode, Pt was employed as the auxiliary electrode, whereas the deposition layer sample formed the working electrode. The test was conducted in an electrolytic solution with 3.5 wt.% of NaCl at 30 °C. Figure 9a provides the Nyquist plots for the deposited layers with 0 wt.% and 0.5 wt.% Y_2_O_3_, and also the corresponding equivalent circuit that was obtained using the ZView software. The addition of 0.5 wt.% Y_2_O_3_ nanoparticles led to enhanced corrosion resistance of the NiCoCr-based superalloy layer that was prepared by the LMD technique. Figure 9b shows the Tafel polarization curves for the deposited layers of NiCoCr-based superalloy with 0 wt.% and 0.5 wt.% Y_2_O_3_. In comparison to the deposited layer with 0 wt.% Y_2_O_3_, the polarization curves for the layer with 0.5 wt.% Y_2_O_3_ were slightly displaced towards the region of lower corrosion current density. The corrosion voltages of the deposited layers with 0 wt.% and 0.5 wt.% Y_2_O_3_ were found to be −0.440 V and −0.323 V, respectively, demonstrating that the addition of Y_2_O_3_ led to significantly enhanced corrosion resistance in the deposited layer of NiCoCr-based superalloy.

Enhancement of corrosion resistance by the addition of Y_2_O_3_ can be attributed to three factors. Firstly, the addition of Y_2_O_3_ led to grain refinement and a homogeneous microstructure in addition to the reduction in defects at the grain boundaries, which greatly prevented the aggregation of corrosive ions at the grain boundaries. Secondly, the introduction of Y_2_O_3_ led to the development of a compact deposition layer, which drastically reduced the gaps and cracks during the deposition process. This study provides sufficient evidence that the introduced particles can occupy existing gaps and cracks, thereby preventing corrosive ions from reaching the substrate. Lastly, the Y_2_O_3_ nanoparticles were uniformly distributed, and their standard potential was more positive than that of the NiCoCr-based superalloy. Hence, a galvanic cell was developed with the NiCoCr-based superalloy matrix acting as the anode and the uniformly distributed Y_2_O_3_ nanoparticles playing the role of the cathode. As a result, the corrosion mechanism was transformed from a local and pitting erosion to a homogeneous erosion [26,27,28,29]. Additionally, as an additional advantage Y_2_O_3_ nanoparticles, it has been also reported that they can prevent the expansion of pits within the corrosion medium by serving as an inert barrier [30].

## 4. Conclusions

(1)NiCoCr-based superalloy layers prepared by LMD were significantly vulnerable to intergranular cracking. Both sides of the crack were observed to be smooth, indicating that the crack was induced by liquid phase of the grain boundaries.(2)The presence of Y_2_O_3_ did not affect the morphology of the columnar dendritic grains present in the NiCoCr-based superalloy prepared by the LMD technique. Instead, it induced a grain refinement effect that hindered growth of the columnar dendritic grains, thus effectively inhibiting crack formation and propagation.(3)The presence of Y_2_O_3_ led to improved mechanical performance of the deposited layer, increasing its tensile strength and elongation by 18.3% and 136%, respectively. The presence of Y_2_O_3_ also led to increased corrosion voltage in the deposited layer of the alloy within the corrosion medium, thereby improving its corrosion resistance.

## Figures and Tables

**Figure 1 materials-16-03616-f001:**
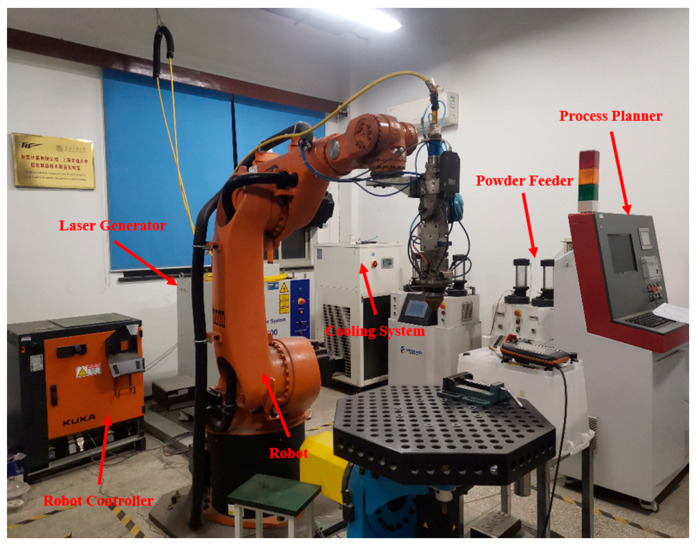
Laser metal deposition system.

**Figure 2 materials-16-03616-f002:**
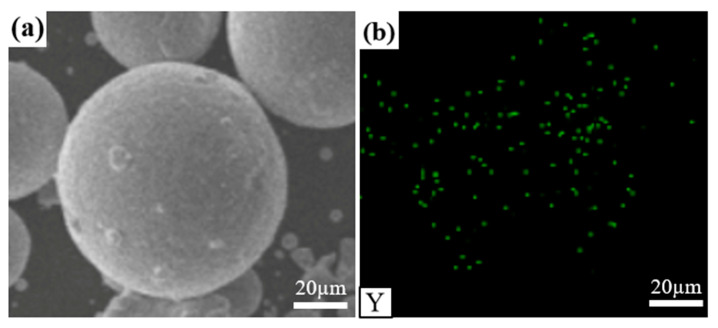
(**a**) SEM images and (**b**) Y-element EDS result of the as-prepared Y_2_O_3_/NiCoCr-based powder.

**Figure 3 materials-16-03616-f003:**
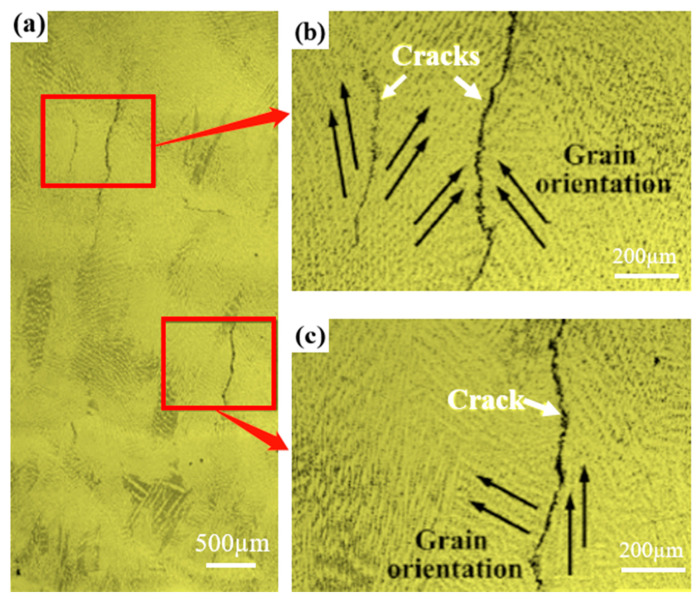
(**a**) Crack distribution, (**b**,**c**) crack morphology of the multi-layer, multi-pass NiCoCr-based superalloy prepared by LMD.

**Figure 4 materials-16-03616-f004:**
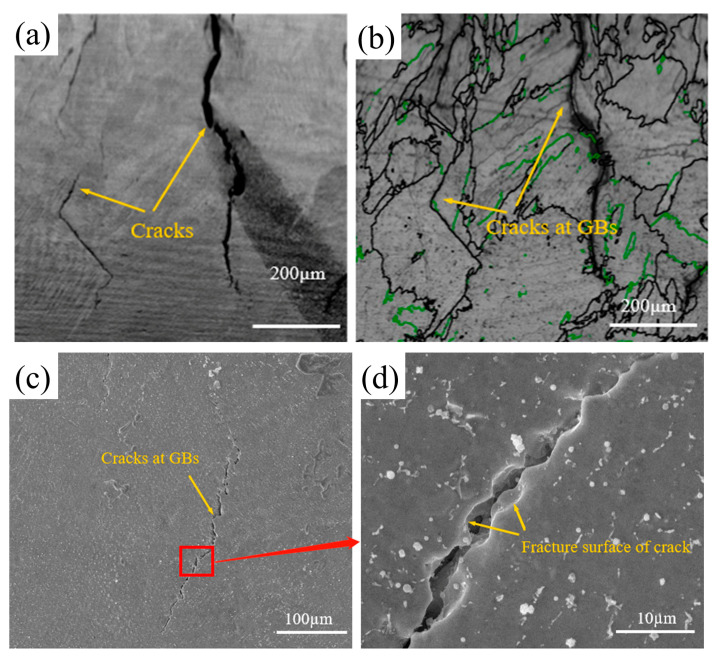
(**a**) Morphology, (**b**) high-angle and low-angle grain boundaries, (**c**) macroscopic morphology and (**d**) local morphology of the crack zone for the NiCoCr-based superalloy prepared by LMD.

**Figure 5 materials-16-03616-f005:**
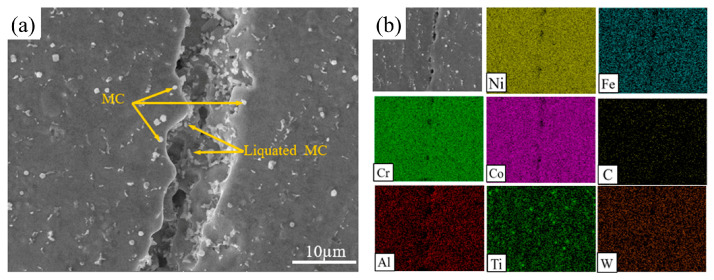
(**a**) SEM image, (**b**) EDS results for the cracks formed in the NiCoCr-based superalloy prepared by LMD.

**Figure 6 materials-16-03616-f006:**
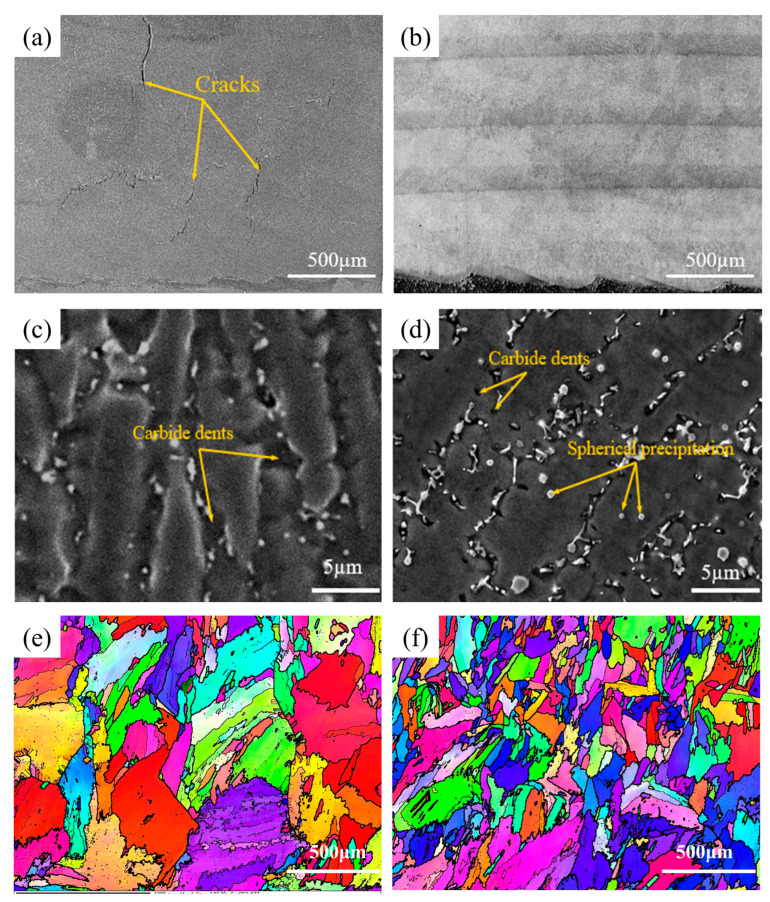
Morphological imaging results along growth direction of the NiCoCr-based superalloy prepared by LMD for 0 wt.% and 0.5 wt.% Y_2_O_3_. (**a**,**c**) and (**b**),**d**) SEM images for 0 wt.% and 0.5 wt.% of Y_2_O_3_, respectively. (**e**) and (**f**) Inverse figure pole (IFP) maps, LMD for 0 wt.% and 0.5 wt.% Y_2_O_3_, respectively.

**Figure 7 materials-16-03616-f007:**
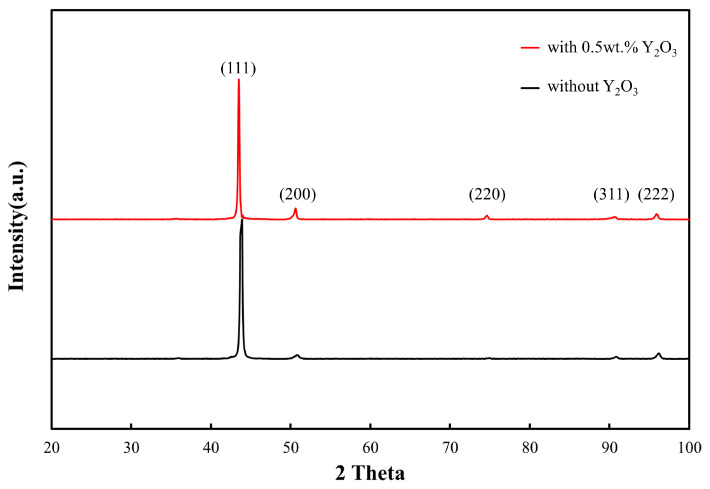
XRD pattern of the NiCoCr-based superalloy prepared by LMD for 0 wt.% and 0.5 wt.% Y_2_O_3_.

**Figure 8 materials-16-03616-f008:**
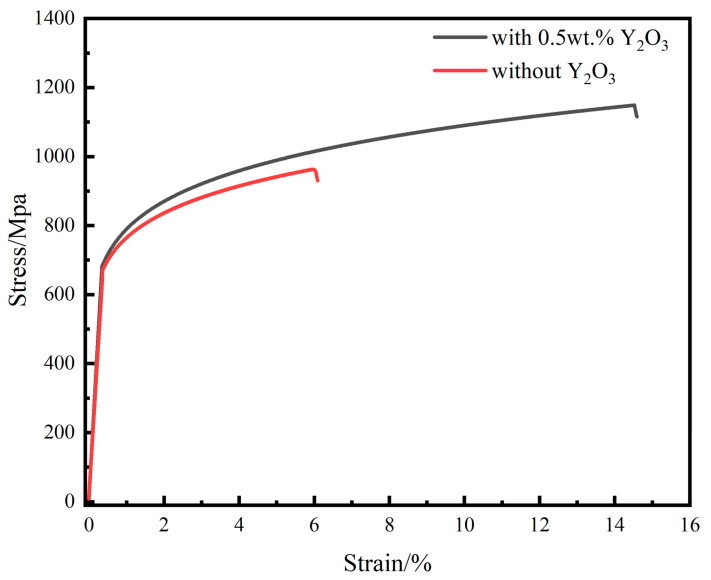
Tensile stress–strain curves for NiCoCr-based superalloy and 0.5wt.% Y_2_O_3_/NiCoCr-based composite.

**Figure 9 materials-16-03616-f009:**
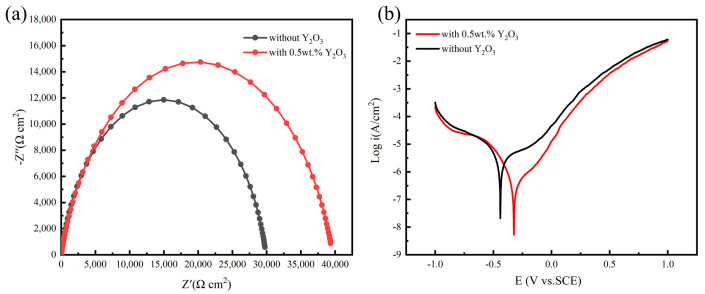
(**a**) Nyquist diagram, (**b**) Tafel polarization curves for 0 wt.% and 0.5 wt.% Y_2_O_3_.

**Table 1 materials-16-03616-t001:** Chemical compositions of NiCoCr-based superalloy powder (wt.%).

Ni	Fe	Co	Cr	W	Al	Ti	C
38	5	20	25	8~9	3	0.5	0.8~0.9

## Data Availability

The authors declare that the data supporting the findings of this study are available within the paper.

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
