# Peer review of "Crack Inhibition and Performance Modification of NiCoCr-Based Superalloy with Y2O3 Nanoparticles by Laser Metal Deposition"

_materials, 2023, doi:10.3390/ma16103616_

Round 1

Reviewer 1 Report

In this study, a precipitation strengthening NiCoCr-based superalloy with favorable wear and corrosion resistance was designed for ultra-supercritical power generation equipment. Y2O3 was added into the alloy powder to reduce or eliminate hot cracks initiated during the manufacturing process of the NiCoCr-based superalloy by advanced additive manufacturing techniques metal deposition. The present results show that adding Y2O3, it refines grains significantly; the increase in grain boundaries makes the residual thermal stress more uniform to reduce the risk of hot cracking. Also, the authors found that the addition of Y2O3 nanoparticles enhanced the ultimate tensile strength of the superalloy compared with original superalloy. Finally, the authors found that the corrosion resistance was also improved. The issue of the paper is attractive due to it has intense attention to use for manufacturing aerogene blades, turbine disks, etc. The present results are consistent.

The paper could be published. But there are a few points that should be attended, which I feel, they would improve the presentation.

Page 2, 80-84 lines. The authors state “As provided in Table 1, Ni, Co, Cr and W exhibit good thermal and wear resistances, whereas Al, Ti, and Nb are used as solution strengthening elements.”. I did not see how the authors from table concluded Ni, Co, Cr and W exhibit good thermal and wear resistances, whereas Al, Ti, and Nb are used as solution strengthening elements, if table 1 only give the chemical composition.

Page 2, 49-51 lines. The authors state “A good understanding of the mechanism behind crack formation within the deposited layers of NiCoCr-based superalloys is of great significance for prevention and control of crack initiation and propagation.”. That statement needs reference.

Page 2, 49-53 lines. The authors state “A good understanding of the mechanism behind crack formation within the deposited layers of NiCoCr-based superalloys is of great significance for prevention and control of crack initiation and propagation. Previous studies devoted to both the microstructure, as well as the crack formation in NiCoCr-based superalloys prepared by LMD have so far been rather inconclusive.”. First the authors mention that there is “A good understanding of the mechanism behind crack formation within the deposited layers of NiCoCr-based superalloys …”, later they indicate that “Previous studies devoted to both the microstructure, as well as the crack formation in NiCoCr-based superalloys prepared by LMD have so far been rather inconclusive.”.

That is not contradictory?   

Author Response

Reviewer 1: In this study, a precipitation strengthening NiCoCr-based superalloy with favorable wear and corrosion resistance was designed for ultra-supercritical power generation equipment. Y2O3 was added into the alloy powder to reduce or eliminate hot cracks initiated during the manufacturing process of the NiCoCr-based superalloy by advanced additive manufacturing techniques metal deposition. The present results show that adding Y2O3, it refines grains significantly; the increase in grain boundaries makes the residual thermal stress more uniform to reduce the risk of hot cracking. Also, the authors found that the addition of Y2O3 nanoparticles enhanced the ultimate tensile strength of the superalloy compared with original superalloy. Finally, the authors found that the corrosion resistance was also improved. The issue of the paper is attractive due to it has intense attention to use for manufacturing aerogene blades, turbine disks, etc. The present results are consistent.

The paper could be published. But there are a few points that should be attended, which I feel, they would improve the presentation.

1.Page 2, 80-84 lines. The authors state “As provided in Table 1, Ni, Co, Cr and W exhibit good thermal and wear resistances, whereas Al, Ti, and Nb are used as solution strengthening elements.”. I did not see how the authors from table concluded Ni, Co, Cr and W exhibit good thermal and wear resistances, whereas Al, Ti, and Nb are used as solution strengthening elements, if table 1 only give the chemical composition.

Response1: Thanks for your suggestion. The less rigorous expression has been modified as below and the relevant references have been added.

‘’As provided in Table 1, the matrix including Ni, Co and Cr exhibits good thermal resistances and mechanical behavior under high temperature with addition of Al, Ti, and Nb, which are used as solution strengthening elements [13, 14]. W is also added to enhance wearing resistance and generate strengthening effect by causing lattice distortion because of its largest atomic radius [15].”

[13] Y. Qi, T. Cao, H. Zong, et al. Enhancement of strength-ductility balance of heavy Ti and Al alloyed FeCoNiCr high-entropy alloys via boron doping, J. Mater. Sci. Tech. 75 (2021) 154–163.

[14] H. Liang, D. Qiaoa, J. Miao, et al. Anomalous microstructure and tribological evaluation of AlCrFeNiW0.2Ti0.5 high-entropy alloy coating manufactured by laser cladding in seawater. J. Mater. Sci. Tech. 85 (2021) 224–234.

[15] T. Wu, Y. Chen, S. Shi, et al. Effects of W Alloying on the Lattice Distortion and Wear Behavior of Laser Cladding AlCoCrFeNiWx High-Entropy Alloy Coatings, Materials 14 (2021) 5450.

  1. Page 2, 49-51 lines. The authors state “A good understanding of the mechanism behind crack formation within the deposited layers of NiCoCr-based superalloys is of great significance for prevention and control of crack initiation and propagation.”. That statement needs reference.

Response2: Thanks for your suggestion. The relevant references have been added.

[6] Sindo Kou, A criterion for cracking during solidification, Acta Mater., 88 (2015) 366–374.

[7] S. Banoth, T. Palleda, S. Shimazu, et al. Yttrium’s effect on the hot cracking and creep properties of a Ni-based superalloy built up by additive manufacturing, Materials 14 (2021) 1143

  1. Page 2, 49-53 lines. The authors state “A good understanding of the mechanism behind crack formation within the deposited layers of NiCoCr-based superalloys is of great significance for prevention and control of crack initiation and propagation. Previous studies devoted to both the microstructure, as well as the crack formation in NiCoCr-based superalloys prepared by LMD have so far been rather inconclusive.”. First the authors mention that there is “A good understanding of the mechanism behind crack formation within the deposited layers of NiCoCr-based superalloys …”, later they indicate that “Previous studies devoted to both the microstructure, as well as the crack formation in NiCoCr-based superalloys prepared by LMD have so far been rather inconclusive.”.

That is not contradictory?  

Response3: Thanks for your suggestion. The seemingly contradictory statement has been replaced by a more appropriate one as below.

In order to prevent the occurrence of hot cracks, exploring the mechanism behind crack formation within the de-posited layers of NiCoCr-based superalloys is of great significance. However, the mechanisms of hot cracking are influenced tremendously by the alloy compositions and processing methods. Therefore, previous studies devoted to both the microstructure, as well as the crack formation in NiCoCr-based superalloys prepared by LMD have so far been rather inconclusive.

Reviewer 2 Report

The manuscript contains interesting data on the effect of using Y2O3 nanoparticles to increase the mechanical and anti-corrosion stability of NiCoCr alloy layers. Research methods are based primarily on imaging techniques, selected mechanical and electrochemical methods. If the research results are to be helpful for engineers, it should be clearly written what the modified alloy layers are intended for. Under what conditions should they work? The abstract also needs to be changed in this direction. A comment on the selection of the sample thickness is required too.

There are no present xrd data for these materials that would allow a better understanding of what happens to the grain structure of the alloy after adding Y2O3. However, this is not a prerequisite for the publication of the manuscript.

The results that are given are correct and deserve to be disseminated. Therefore, after a little editing, the manuscript can be published.

Author Response

Reviewer 2: The manuscript contains interesting data on the effect of using Y2O3 nanoparticles to increase the mechanical and anti-corrosion stability of NiCoCr alloy layers. Research methods are based primarily on imaging techniques, selected mechanical and electrochemical methods.

  1. If the research results are to be helpful for engineers, it should be clearly written what the modified alloy layers are intended for. Under what conditions should they work? The abstract also needs to be changed in this direction. A comment on the selection of the sample thickness is required too.

Response1: Thanks for your suggestion. The application background of alloy layer has been added to abstract and introduction.

(1) In abstract: “A new precipitation strengthening NiCoCr-based superalloy with favorable mechanical performance and corrosion resistance was designed for ultra-supercritical power generation equipment. The degradation of mechanical properties and steam corrosion at high temperature put forward higher requirements for alternative alloy materials.

(2) In introduction, “A new kind of NiCoCr-based superalloy has been developed for ultra-supercritical power equipment at temperatures above 650 °C. The degradation of mechanical properties and steam corrosion at high temperature put forward higher requirements for alternative alloy materials. As a precipitated strengthening alloy, it can maintain considerable mechanical properties at high temperature. In addition, the high content of Cr also gives the alloy excellent oxidation and corrosion resistance. Therefore, it is considered as a suitable candidate material for ultra-supercritical power generation equipment.

(3) In this research, the sample thickness is taken by 3 mm. In fact, the sample thickness is decided by the detailed application. Benefit from the flexibility of DED, the thickness of the sample can be adjusted as required.

2.There are no present xrd data for these materials that would allow a better understanding of what happens to the grain structure of the alloy after adding Y2O3. However, this is not a prerequisite for the publication of the manuscript. The results that are given are correct and deserve to be disseminated. Therefore, after a little editing, the manuscript can be published.

Response2: Thanks for your suggestion. The XRD data and the relevant description are appended in the manuscript.

‘’The XRD patterns were also compared in Fig. 7. And it can be seen that both samples have typical FCC characteristics. With addition of Y2O3, the intensity of grain orientations (200) and (220) are enhanced, which may be related to grain refinement, resulting in a slight increase in anisotropy. However, Because of the low content of Y2O3 (0.5 wt.%) and the strong tendency of epitaxial growth of NiCoCr-based superalloy, there is no significant difference between the XRD patterns.’’
